# The impact of news exposure on collective attention in the United States during the 2016 Zika epidemic

**Michele Tizzoni**[ID]*, **André Panisson**[ID], **Daniela Paolotti**[ID], **Ciro Cattuto**

ISI Foundation, Turin, Italy

* michele.tizzoni@isi.it

**Data Availability Statement:** Data are available from the Zenodo repository (https://doi.org/10.5281/zenodo.3603916).

## Abstract

In recent years, many studies have drawn attention to the important role of collective awareness and human behaviour during epidemic outbreaks. A number of modelling efforts have investigated the interaction between the disease transmission dynamics and human behaviour change mediated by news coverage and by information spreading in the population. Yet, given the scarcity of data on public awareness during an epidemic, few studies have relied on empirical data. Here, we use fine-grained, geo-referenced data from three online sources—Wikipedia, the GDELT Project and the Internet Archive—to quantify population-scale information seeking about the 2016 Zika virus epidemic in the U.S., explicitly linking such behavioural signal to epidemiological data. Geo-localized Wikipedia pageview data reveal that visiting patterns of Zika-related pages in Wikipedia were highly synchronized across the United States and largely explained by exposure to national television broadcast. Contrary to the assumption of some theoretical epidemic models, news volume and Wikipedia visiting patterns were not significantly correlated with the magnitude or the extent of the epidemic. Attention to Zika, in terms of Zika-related Wikipedia pageviews, was high at the beginning of the outbreak, when public health agencies raised an international alert and triggered media coverage, but subsequently exhibited an activity profile that suggests nonlinear dependencies and memory effects in the relation between information seeking, media pressure, and disease dynamics. This calls for a new and more general modelling framework to describe the interaction between media exposure, public awareness and disease dynamics during epidemic outbreaks.

## Author summary

Despite its importance for public health policy-makers, understanding the impact of media coverage on collective attention during disease outbreaks remains an elusive research task, due to the lack of available data, especially at high spatial granularity. In this paper, we study the dynamics of collective attention received by the 2016 Zika epidemic in the USA and its interplay with the media coverage of the outbreak, at level of US states and cities. We measure the attention to Zika through geo-localized Wikipedia page view

**Funding:** MT, AP, DP and CC acknowledge the support by the Lagrange Project of the ISI Foundation funded by the CRT Foundation. The funders had no role in study design, data collection and analysis, decision to publish, or preparation of the manuscript.

**Competing interests:** The authors have declared that no competing interests exist.

data, and we compare it with mentions of Zika in US news outlets and TV shows. We also compare the collective attention received by the outbreak with the incidence of Zika reported by the US Centers for Disease Control and Prevention in each state. We find that the attention dynamics was highly synchronized across states, irrespective of the local risk of transmission of the virus. By building a linear regression model, we show that the dynamics of collective attention is highly predictable, even at state level, only based on the national media coverage received by the outbreak.

## Introduction

The advent of the digital era has radically changed the way individuals search for information and this is particularly relevant for health-related information [1]. A 2013 study [2] found that 59% of U.S. adults had looked for health information on the Web in the previous year and that about one in three U.S. adults use the Internet to figure out what medical condition they have. The fruition of news sources, either traditional such as television, radio and newspapers, or digital such as Web news or online social networks, has become crucial in how health information is delivered and it can play a fundamental role in shaping opinions, awareness and behaviours. In the past ten years, several studies have addressed the impact of awareness and information spread during epidemic outbreaks and it has been reported that the degree of public attention and concern induced by an epidemic threat might affect the disease transmission dynamics [3–7]. However, modeling efforts have been mostly theoretical and a large-scale empirical characterization of the adoption of health protective behaviours, either induced by information spread or media exposure, and its interplay with the disease dynamics during an epidemic outbreak has been elusive so far due to the lack of available data [8].

Here, we study a large-scale dataset on spatio-temporally resolved accesses to Wikipedia pages on the 2015-2016 Zika virus (ZIKV) epidemic, regarded as a proxy for collective attention to this emerging health threat. The epidemic started in Brazil in 2015 and spread to other parts of South and North America in 2016. This study focuses on attention patterns in the United States throughout 2016, and on their relation to media coverage of the epidemic.

ZIKV is a RNA virus from the *Flaviviridae* family which is mainly transmitted by infected *Aedes* mosquitoes, although there have been cases of sexual and perinatal transmission. Infection is mostly asymptomatic or associated with mild symptoms [9] but it can lead to serious and sometimes fatal neurological defects in neonates born to ZIKV infected women. In particular, following the association between ZIKV and a cluster of microcephaly cases in Brazil [10], the World Health Organization (WHO) declared the ZIKV epidemic a Public Health Emergency of International Concern (PHEIC) on February 1st, 2016 [11]. The emergency lasted until November 18th 2016, when the WHO declared the PHEIC to be over [12]. As of March 2017, ZIKV has spread worldwide to 79 countries where there has been evidence of an ongoing vector-borne virus transmission. The most affected region has been the American continent with 47 countries or territories reporting local ZIKV transmission, due to the extensive presence of *Aedes* mosquitoes in almost all the region's countries [13]. In such epidemiological context, the ZIKV epidemic has posed peculiar communication challenges to the public due to its association with microcephaly in newborns, its transmission modalities, and its prevalence in areas where the virus was never detected before and that was suddenly characterized by intense international travel due to the 2016 Summer Olympics [13–15].

The communication challenges posed by ZIKV are well exemplified in a manual released in 2017 by the WHO European Region [16] on how to deal with the complex communication

challenges of ZIKV and mosquito borne disease outbreaks in general. Among the main communication issues listed in the manual, besides *Dealing with uncertainty* and *Rumor management*, there are *Increase in information demand* and *Managing evolving information*. The last two are particularly interesting as they refer to the massive demand for information from the public, and from media as well, and to the necessity of adapting public health communication as more is learned about the outbreak and scientific knowledge about Zika virus and its complications evolve.

Public polls conducted in the United States evidenced the lack of knowledge about ZIKV in the general population and more specifically in groups at risk, such as pregnant women [17]. The novelty of the disease and the lack of previous knowledge of it in the affected areas make the 2016 ZIKV epidemic an ideal case study to characterize collective attention patterns, identify their drivers and test traditional modeling assumptions. Intuitively, mass media coverage represents the main driver of public attention during an epidemic. Indeed, several peculiarities of media narratives around public health hazards [18] and infectious diseases [19] have been elucidated, but a general and quantitative comprehension of how the public opinion responds to media exposure during an emerging epidemic threat is still lacking. The majority of modeling studies assume that media exposure is driving behavioural changes, hence media exposure effects are incorporated into some kind of *media function* that modulates individual behaviors and may affect disease dynamics [20–22]. The general assumption is that as the number of cases increases and is reported by mass media, the susceptibility of individuals will decrease due to increasing awareness and the associated behavioral changes [21]. However, for most disease outbreaks, such modeling assumptions have never been supported by direct empirical evidence.

Our study analyses time-resolved and geo-localized Wikipedia pageview counts to investigate the dynamics of public attention in the United States, during the 2016 ZIKV epidemic. Accesses to Wikipedia pages represent a signal of information seeking behavior, defined as the deliberate process in which individuals actively aim to acquire new knowledge by searching for information on a specific topic [23]. In our study, we considered such indicator to be a proxy for measuring the collective attention to the outbreak. Specifically, we considered the daily pageview counts on 128 different Zika-related Wikipedia articles (96 languages) in the U.S. to be an unambiguous indicator of collective attention to the epidemic. We investigated the temporal and spatial patterns of pageviews in relation to the timeline of ZIKV incidence reported by the US Centers for Disease Control and Prevention (CDC), and in relation to the coverage of the ZIKV epidemic by local and national media sources. In particular, we focused on news coverage of the ZIKV epidemic by online media and television in 2016, available in digital format through the GDELT project and the Internet Archive (see Methods for a full description of the data under study).

## Results

### Temporal profile of collective attention and news coverage

Public attention and media coverage of the 2016 ZIKV epidemic showed a distinct and synchronous temporal pattern, as seen in Fig 1. The daily timeline of Wikipedia pageviews (Fig 1A) highlights two distinct peaks of attention in 2016: the first, in the beginning of February 2016, corresponding to the international alert raised by the WHO and by the CDC at national level; the second, in August 2016, corresponding to the Summer Olympics in Rio de Janeiro, which attracted significant attention due to the health concerns for athletes and the related risk of case importation. Such spikes of attention visibly correspond to similar spikes in the media coverage profiles, both in the TV coverage of the epidemic (Fig 1B) and in the Web news

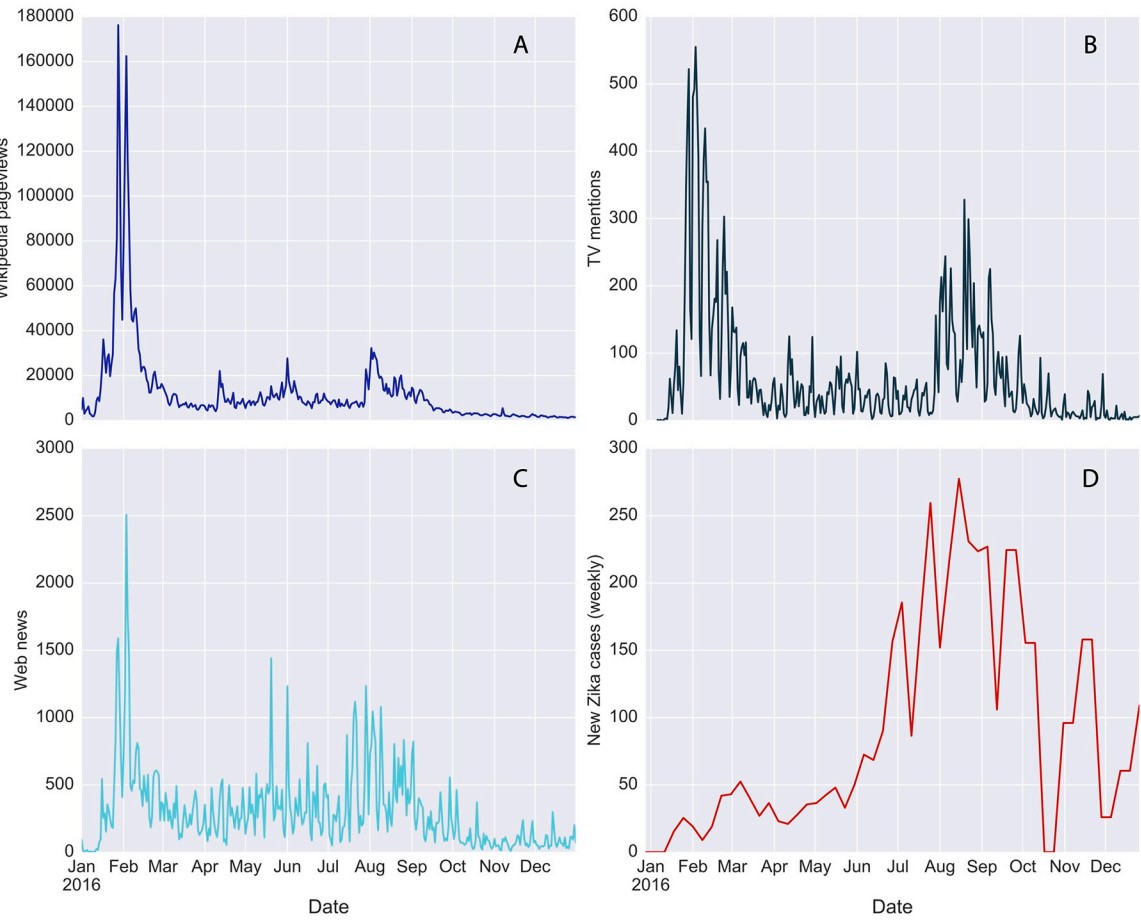

**Fig 1. Attention, media coverage and disease incidence of the Zika virus in the USA in 2016.** (A) Daily Wikipedia pageview counts of Zika related pages. **B**, Daily mentions of the word "Zika" in TV programs broadcasted in the U.S. extracted from the TV Internet Archive. **C**, Daily number of Web news mentioning "Zika" extracted from the GDELT project. **D**, Weekly incidence of the Zika virus reported by the CDC. Originally reported case counts were smoothed with a biweekly rolling average.

coverage of Zika (Fig 1C). The three time series are indeed highly correlated, with a Pearson's correlation coefficient $r = 0.74$, ($p < 10^{-4}$), for the Wikipedia pageviews and the Web news time series, and $r = 0.80$, ($p < 10^{-4}$), for the Wikipedia attention and the TV coverage (see S2 Table in the Supporting Information).

On the contrary, while the profile of Wikipedia pageviews shows a temporal pattern that is very similar to the one displayed by mentions of Zika in media outlets, the temporal profile of the disease incidence is qualitatively very different. The number of new ZIKV cases in the United States reported by the CDC every week (Fig 1D) gradually increased from the beginning of 2016 until the summer, with a peak between the end of August and the beginning of September. Notifications of new cases declined afterwards. Summer 2016 was also characterized by the first reports of local ZIKV transmission in Florida and in Texas, events that were also responsible for an increased news coverage of Zika. However, the surge of reported ZIKV cases in the United States did not result in an increased level of attention with respect to the initial spike observed at the beginning of the outbreak. The epidemic profile is indeed not correlated with the timeline of Wikipedia pageviews ($r = −0.15$, $p = 0.26$) or the media coverage profiles ($r = 0.04$, $p = 0.80$ for TV and $r = 0.10$, $p = 0.47$, for Web news). Such dynamics of public attention can be ascribed to the initial novelty of the outbreak, which was presented as a

novel and serious health threat even in the presence of a small number of imported cases. As the extension of the outbreak and the associated risks became clearer to the public, the interest of Americans in looking for additional information on Wikipedia faded over the course of the year, with relatively smaller increments linked to important events such as the Olympics. Such observation is consistent with the presence of a memory effect in the dynamics of Wikipedia pageviews: individuals retain information for some time before their attention toward a topic is elicited again by novel events or anniversaries [24, 25].

## Spatial patterns of collective attention

The available spatial granularity of Wikipedia pageview data allowed us to further inspect how the above picture changes when moving from a national perspective to States and U.S. cities. Notably, the temporal dynamics of attention to the Zika-related Wikipedia pages in 2016 was highly synchronized across all the 50 States. Although the relative risk of case importation and local transmission varied significantly from state to state, being the Southern States more at risk due to vector's presence and abundance [26], the Wikipedia pageview timelines were all highly correlated, as shown in Fig 2. The Pearson correlation coefficient of the cross-correlation matrix of Wikipedia pageview time series by State ranges from $r = 0.77$ ($p < 10^{-4}$) for Delaware and Montana, to $r = 0.99$ ($p < 10^{-4}$) for New York and New Jersey. Overall, the correlation of the Wikipedia pageviews in each state with the national timeline was always higher than $r = 0.88$ ($p < 10^{-4}$), indicating a high degree of spatial uniformity across the country. Given the above mentioned correlation of Wikipedia pageviews with the TV coverage of the epidemic and the mentions of Zika on the Web, the attention patterns at State level were also highly correlated with the national media coverage suggesting a fundamental role of news exposure as a driver of public attention at all geographic scales.

One could argue that local patterns of attention may be influenced by local news and local epidemic events, such as case importations or a local increase of disease prevalence. We tested these hypotheses by comparing Wikipedia pageview counts in each state to Web news mentioning the word "Zika" and the name of the state, and to the local ZIKV incidence profiles. Attention profiles in each state were generally positively correlated to Web news mentioning the name of the state, however the degree of correlation ranged from $r = 0.004$ ($p = 0.98$) in Wisconsin to $r = 0.75$ ($p < 10^{-4}$) in Texas, showing significant spatial differences across the country (see S3 Table in the Supporting Information). Interestingly, ZIKV incidence in each state could explain such geographic variations as a negative driver of attention. On the one hand, local patterns of attention in each state were generally not correlated with disease incidence, with the exception of Montana ($r = 0.33$, $p = 0.02$), as shown in S4 Table of the Supporting Information. On the other hand, Web news covering Zika in each state were positively correlated with the local incidence profiles ($r > 0.20$) only in 13 states out of 50 (see S5 Table in the Supporting Information) and, at the same time, these states showed the smallest degree of correlation between news and attention. A direct comparison of the 50 states ranked by degree of correlation between news and ZIKV incidence, and between news and attention, showed a negative rank correlation: weighted Kendall's $\tau = -0.25$ [27]. Overall, in those states where local news were following more closely the local epidemic patterns, the dynamics of public attention was not driven much by news. Instead, local attention patterns followed more closely the state news where the latter was more similar to the national one and less correlated with the local ZIKV epidemiology.

It is natural to ask whether correlations between patterns of attention and disease risk may change by looking at different spatial resolutions. To answer this question, we examined the attention to ZIKV in 788 cities of the United States with a population larger than 40,000 and

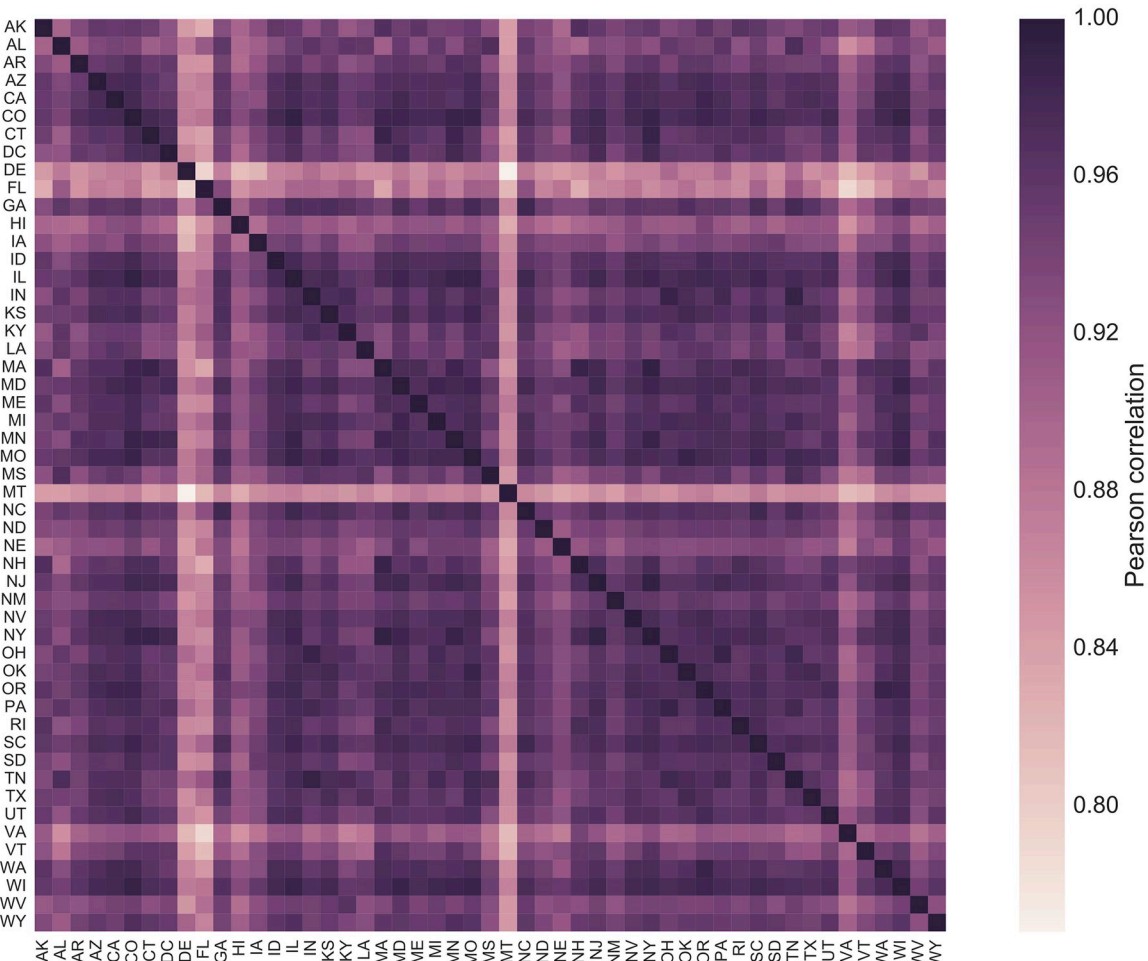

**Fig 2. Correlation of public attention timelines by state.** Pearson correlation matrix of the daily Wikipedia pageviews time series of the 50 states and the District of Columbia.

compared it to their total Wikipedia viewership. By ranking the U.S. cities based on their total volume of Wikipedia pageviews in 2016, and comparing such ranking with the one based on pageviews of Zika-related articles only, we identified locations where the attention to ZIKV was higher than expected. As shown in Fig 3A, cities on the East Coast of Florida showed the highest relative attention to ZIKV, when compared to their overall Wikipedia activity. Other relevant outliers with high attention were cities in Texas and in the Northeast. On the contrary, the lowest attention to ZIKV was observed in cities in California, and in the Midwest (Fig 3B). These results suggest that increases in public attention at city level may be explained by risk perception due to the presence of the vector (as in Florida and Texas). However, the high level of attention in other places, such as Union City, NJ, can not be easily explained by epidemiological risk factors and it may be due to specific events, such as one or more case importations, that do not appear in our dataset.

## Time series analysis

The correlation analysis based on the Pearson's coefficient could be influenced by autocorrelations in both the dependent and independent variables under consideration. To better assess

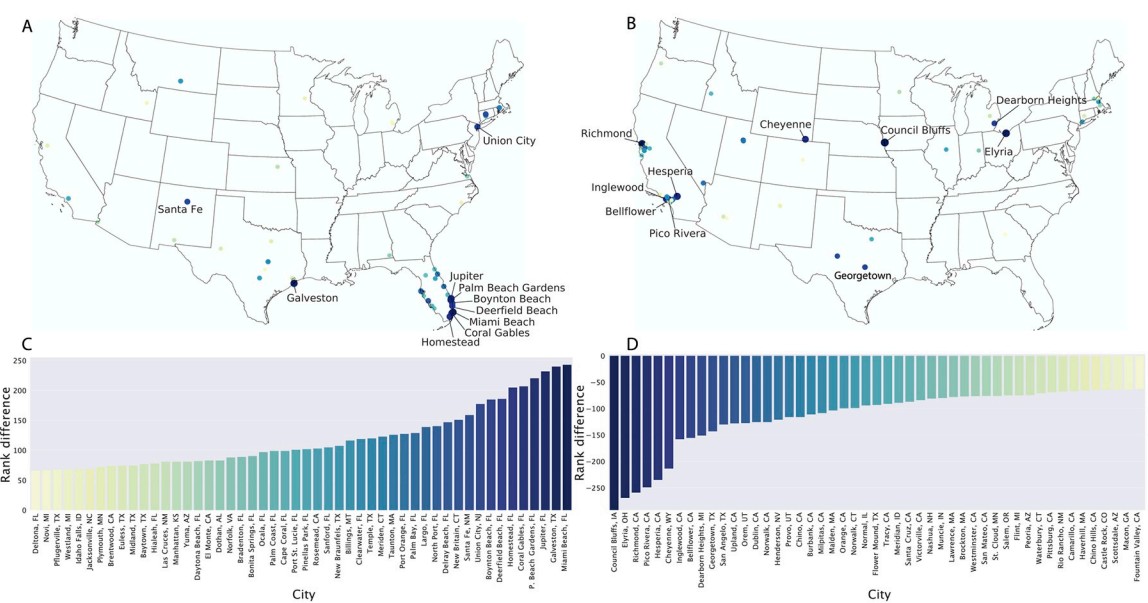

**Fig 3. Spatial patterns of attention.** Cities of the United States, with population higher than 40,000, where the volume of attention to ZIKV related pages was higher (panel A) or lower (panel B) than expected based on the total volume of pageviews to Wikipedia in 2016. The maps only show the 50 cities with the largest positive (panel C) or negative (panel D) difference in their pageview rankings, based on ZIKV related pages and the full Wikipedia. The labels on the maps highlight the 10 cities with highest (panel A) or lowest attention (panel B).

the mutual relationships between news coverage, Wikipedia pageviews and Zika incidence, we turn to a Vector Autoregression (VAR) model, a technique that is commonly used in the analysis of multivariate time series [28]. In a VAR($L$) model, each variable can have an influence on all other variables with a maximum time lag equal to $L$ (see Materials and methods).

First, we build a VAR model using daily time series of Web news, TV captions and Wikipedia pageviews, through the VARS package in R [29]. By computing the Schwartz-Bayes information criterion (BIC) and the Hannan-Quinn criterion for different VAR($L$) models, with $L$ ranging between 1 and 40, we identify the optimal time lag to be $L = 8$ days. A Granger causality test based on the best VAR model shows that each variable does Granger-cause the other two variables in the model, with a smaller effect of Web news ($p < 0.05$) with respect to TV ($p < 10^{-4}$) and Wikipedia pageviews ($p < 10^{-4}$). On the other hand, a Wald-type test supports a relationship of instantaneous causality among all the three variables: Web news instantaneously cause Wikipedia pageviews and TV mentions ($\chi^2 = 86.321$, $df = 2$, $p < 10^{-6}$); TV mentions instantaneously cause Wikipedia pageviews and Web news ($\chi^2 = 97.335$, df = 2, $p < 10^{-6}$); Wikipedia pageviews instantaneously cause Web news and TV mentions ($\chi^2 = 89.23$, $df = 2$, $p < 10^{-6}$).

To include the ZIKV incidence timeline in our analysis, we build another VAR model with 4 weekly time series: Zika-related pageviews, TV mentions, Web news and ZIKV incidence. For this model we identify the optimal time lag to be $L = 2$ weeks, based on the Schwartz-Bayes information criterion. A Granger causality test using the VAR(2) model identifies a causal relationship between Wikipedia pageviews and the other variables ($p < 10^{-3}$), and a causal relationship between Web news and the other variables ($p < 0.05$). Again, based on a Wald test, an instantaneous causal relationship between TV captions, Web news and Wikipedia pageviews and the other variables is supported ($p < 10^{-3}$). ZIKV incidence, instead, does not Granger-

cause the other time series nor there is an instantaneous causal relationship between the epidemic profile and the other variables ($p$ = 0.27).

While a preferential causal direction between media coverage and Wikipedia pageviews does not emerge from the Granger causality analysis, our results support the idea that the media signal and the pageview timeline are highly synchronized, so that the hypothesis of instantaneous causality is supported, both at daily and weekly scale. On the contrary, the epidemiological ZIKV curve does not show to have a predictive power nor in a Granger-causal framework nor as an instantaneous driver of the public attention.

## An equal-time predictive model of collective attention

Prompted by the results of the time series analysis, we consider the following task: now-casting the number of Zika-related Wikipedia pageviews in each state based on the volume of nation-wide media coverage.

We begin by building an equal-time regression model that predicts the weekly number of Zika-related Wikipedia pageviews for each state, rescaled by state population, based exclusively on the frequencies of Zika-related mentions in Web news and TV closed captions. That is, we assume that information seeking behavior in Wikipedia is driven, at any given point in time, by same-week exposure to media sources. Since our goal is uncovering drivers of collective attention, rather than achieving optimal prediction of the empirical time series, we choose an equal-time modeling approach over standard time series modeling techniques (e.g., autoregressive models). More specifically, we start with a linear regression model that predicts population-rescaled pageview counts for a given week and a given state using only national Web news and TV data for the same week. We focus on 43 states with population in excess of 1 million, comprising more than 98% of the U.S. population according to 2016 United States Census Bureau estimates [30].

These states are also those where the epidemic ZIKV activity in 2016 was the highest. In S7 Table, we report results including all states with the only exception of Alaska, where no ZIKV cases were reported in 2016. We train the model via state-wise cross-validation and evaluate its performance using the determination coefficient $R^2$, the Pearson's correlation coefficient $r$ and the Spearman's $\rho$. Despite its simplicity, this equal-time linear regression demonstrates that both media signals, taken independently, are already quite informative of the Zika population-rescaled pageview time series: using exclusively TV close captions we obtain $R^2$ = 0.61 and $r$ = 0.80, while using only Web news we obtain $R^2$ = 0.52 and $r$ = 0.78. Combining both features, the linear model achieves $R^2$ = 0.63 and $r$ = 0.82. Similar results are obtained when considering the Spearman's $\rho$ as a measure of performance (see Table 1). As model performance is evaluated via state-wise cross-validation, these results highlight that national-level media signals are highly informative of state-level pageview time series, once they are rescaled to take into account population size. As a reference, we compare the results obtained with media features with an equal-time linear regression informed only by the epidemiological signal, that is the ZIKV incidence in each state. As shown in Table 1, first row, the predictive performance of ZIKV incidence is generally poor with $R^2$ = −0.398, $r$ = −0.032 and $\rho$ = 0.152, in line with the time series analysis presented above.

To take into account the possibility of memory effects in the response to media exposure, we enrich the feature space of the regression model with additional features (time series) obtained by filtering the Web news and TV time series with an exponential memory kernel (see Methods for a complete description). The characteristic time $\tau$ of the memory kernel, describing news persistence in the attention response, is a new hyper-parameter of the model to be set via cross-validation. Table 1 summarizes the performance of the model, in terms of

**Table 1. Comparison of model performance with 10 different feature combinations.** For each feature, the average $R^2$, Pearson $r$, Spearman $\rho$ and AIC, computed over 43 states are reported. Average values of $R^2$, $r$ and $\rho$ are computed under K-fold cross-validation ($k = 10$). The standard deviation is reported in parenthesis. AIC is computed for the best set of parameters for each model. The last column reports $\Delta_i = AIC_i − AIC_{min}$ for model comparison.

| Features | $R^2$ | Pearson $r$ | Spearman $\rho$ | AIC | $\Delta_i$ |
|---|---|---|---|---|---|
| ZIKV | -0.398 (0.032) | -0.032 (0.030) | 0.1515 (0.055) | -917.21 | 131.35 |
| TV | 0.6091 (0.040) | 0.8011 (0.021) | 0.7833 (0.030) | -1001.90 | 46.67 |
| Web | 0.5244 (0.046) | 0.7780 (0.026) | 0.7886 (0.030) | -986.01 | 62.54 |
| TV, Web | 0.6266 (0.044) | 0.8209 (0.023) | 0.8105 (0.030) | -1003.92 | 44.65 |
| TV, m(TV) | 0.7062 (0.047) | 0.8675 (0.025) | 0.7677 (0.029) | -1025.90 | 22.66 |
| Web, m(Web) | 0.6352 (0.051) | 0.8178 (0.027) | 0.7419 (0.029) | -1004.53 | 44.03 |
| TV, Web, m(TV) | 0.7318 (0.052) | 0.8761 (0.026) | 0.7963 (0.031) | -1033.91 | 14.64 |
| TV, Web, m(Web) | 0.7603 (0.050) | 0.8942 (0.026) | 0.7546 (0.026) | -1045.90 | 2.66 |
| TV, Web, m(TV), m(Web) | 0.7602 (0.050) | 0.8942 (0.026) | 0.7860 (0.028) | -1044.64 | 3.92 |
| TV, Web, m(TV), m(Web), state_news | 0.7638 (0.045) | 0.8937 (0.024) | 0.7868 (0.027) | -1048.56 | 0.0 |

determination coefficient $R^2$, Pearson's $r$, and Spearman's $\rho$, for 10 different sets of features. The introduction of a memory kernel increases the determination coefficient by about 20%, reaching an average $R^2 = 0.76$. This is obtained for a characteristic time scale $\tau$ of about 2 weeks, over which collective attention is affected by past media exposure.

We also considered state-level news features obtained by counting the weekly number of mentions of each state in Web news. However, adding these features does not significantly improve the model predictions (Table 1, bottom row), although it yields the best performance according to the Akaike Information Criterion (AIC). Overall, by computing the AIC for each model and averaging over all states, three linear models based on TV, Web news, and state news, can be considered equally likely, assuming evidence for $\Delta_i = AIC_i − AIC_{min} < 4$.

## Discussion

Our study demonstrates that the temporal dynamics of Wikipedia pageviews in the United States during the ZIKV 2016 epidemic was highly predictable, even at state level, based on the volume of national and international news sources mentioning Zika and the United States. Collective attention to the ZIKV outbreak thus seems to have been mainly driven by news exposure and much less by the disease transmission dynamics, although the epidemic profile of ZIKV infections varied significantly from state to state and the risk of local transmission was not uniform across the country. Such picture describes a scenario where the awareness of the epidemic in the country is globally present, while local effects, as those due to the local spreading of awareness, play a less important role, following the terminology first introduced by Funk et al. [31].

Media outlets in the U.S. have a prominent role in defining the on-line public discourse [32]. The impact of media exposure on the collective awareness and risk perception during epidemic outbreaks has been investigated in previous works [18, 19, 33], however, only a few studies have attempted to quantitatively measure the effect of media engagement on epidemic awareness using empirical data from Web sources on a large scale [22, 34, 35]. While previous studies have focused on newspaper coverage of epidemics [36], we investigated the relationship between the exposure to TV coverage and online news, and the attention to Wikipedia pages. Our study confirms the high sensitivity of Wikipedia searches to breaking news and official announcements, in particular in the case of disaster events, as found by previous studies [37, 38]. On the other hand, the temporal dynamics of Wikipedia pageviews during the 2016 ZIKV epidemic showed a nonlinear dependence with media coverage: the Wikipedia

pages activity was high in the initial phases of the outbreak, but it declined more quickly than media coverage. This can be explained by the fact that information on Wikipedia is rather static, and users will view Wikipedia pages immediately after the news breaks but they will not return in the next days, unless more recent events renew their attention [37].

Although assuming that collective attention follows media coverage is rather plausible, a Granger-causality analysis did not evidence a preferential causal relationship between page-views and news sources. On the other hand, our data support an instantaneous causal relationship between Wikipedia readership and media coverage, suggesting that nowcasting Wikipedia pageviews based on the volume of media coverage is feasible.

During the 2016 ZIKV outbreak, different aspects of the epidemic, and the risks associated with the disease, received different levels of coverage by traditional media sources, depending on their level of newsworthiness. In general, media coverage was influenced by several factors, and only to a lesser extent by the progression of the epidemic. Although from a journalist's perspective, it may be expected that news did not necessarily follow the number of ZIKV cases, such result highlight the limitations of several behavioral epidemic models that incorporate media effects. Indeed, choices made by journalists regarding newsworthiness of specific aspects related to ZIKV infection affected the level of knowledge and familiarity in the US population [39]. Also, the media narrative around ZIKV evolved over time, as found in a recent study by Yotam Ophir [40]. The content of the ZIKV news coverage shifted from focusing on scientific themes, at the beginning of the outbreak, to the description of social disruptions during the Summer Olympics. Since we did not examine the textual content of news items in our dataset, further research should identify specific themes emerging from media outlets that are most significantly associated to Wikipedia viewership during epidemic outbreaks. Also, we relied on the automatic tagging system provided by GDELT to identify news items mentioning Zika. However, the classification algorithm may have limitations and some items could be less relevant than others. Further research should assess the quality of the GDELT news tagging algorithm in the specific case of epidemic outbreaks [41]. Another limitation of our analysis is that we considered the content of mass media outlets only, and did not look into other news sources, such as online social networks. However, social media often provide an amplification channel for traditional media sources and online users consume information that is not very different from what appears on mass meda [42].

From an epidemiological standpoint, our results are consistent with the recent findings of Bragazzi et al. [43], who analysed various data streams to measure the global reaction to the 2015-2016 ZIKV outbreaks in different countries. Similarly, we did not find any statistically significant correlation between the viewership of Wikipedia pages and the ZIKV incidence data in the U.S. The correlation between ZIKV incidence and media coverage was also mild, and varied from state to state, confirming that media coverage was only relatively influenced by the actual progression of the epidemic over time.

The spatial granularity of Wikipedia pageview data would in principle allow for a more detailed geo-spatial analysis of time series data, in particular regarding languages and behavioral response by geographic areas. On the other hand, increasing the spatial resolution of the pageview data analysis could expose sensitive information about Wikipedia users' locations and preferences. In our study, we decided to limit the risks associated to such analysis by looking only at cumulative pageview data in the US cities.

Our results have implications from a public health perspective. The importance of mass media coverage in eliciting the public attention to announcements by the CDC at the beginning of the Zika outbreak was highlighted by a recent study by Southwell et al. [44]. Consistently, we found that Wikipedia page activity was highest in conjunction of the alert raised by

the CDC, suggesting that media coverage of official communications by public health authorities could effectively capture the public attention and elicit information seeking.

One might argue that our results may not generalize to all epidemic outbreaks. Indeed, the peculiar characteristics of the ZIKV infection, such as its association to mild symptoms and the relatively small size of the population at risk, due to the spatial distribution of the vector, may have influenced the attention dynamics during the outbreak.

As originally defined by Sandman [45], individual perception of risk is the result of a combination between the actual hazard and the emotional response in terms of concern, fear or anger. Sandman's theory is often exemplified by the equation: Risk = Hazard + Outrage and it was one of the first attempt at ascribing to the public a role in risk communication, since in his view it's the interplay between the external threats and personal aspects that shape the meaning of risk. Individual perception of risk during the 2016 ZIKV outbreak has probably decreased quickly, after it became clear that the infection did not pose an immediate threat to most individuals.

Moreover, survey based studies have found the perceived risk of infection for oneself to decrease considerably over time, during the course of a chikungunya outbreak, another vector-borne disease similar to ZIKV [46].

From the point of view of media coverage, the fact that most of ZIKV cases were asymptomatic, may have reduced newsworthiness, as media tend to pay more attention to visually compelling topics [39].

Epidemic outbreaks caused by different pathogens, possibly characterized by a higher transmissibility and more evident clinical symptoms, such as the Ebola virus or pandemic influenza, may lead to different attention patterns by media outlets, raising substantial more concern, and more persistently, in the population. However, it is reasonable to believe that media coverage would be, in any case, the main driver of collective attention, as it also has been during the 2014 West African Ebola virus epidemic [34, 47].

The increasing availability of novel data streams, such as social media, Web search queries and participatory surveillance data, provides an invaluable resource to measure and quantify the complex interplay between the spread of information, collective attention and the epidemiology of infectious diseases [48, 49]. Recently, Wikipedia pageview data have been increasingly used by researchers in epidemiology and infectious disease modeling [50, 51]. The overall value of Wikipedia data to measure and forecast the dynamics of infectious diseases has been debated [52] and, in general, Wikipedia-based forecasting models have been proved successful in the case of endemic or seasonal diseases, such as influenza, dengue or tubercolosis [51]. On the other hand, our study demonstrates that Wikipedia page viewership can provide a temporally resolved measure of collective attention during epidemic outbreaks caused by novel emerging diseases, at a high spatial granularity. Previous works have investigated the effects of external events on the activity of Wikipedia editors and on the number of pageviews [53, 54]. More generally, the characterization of the usage of Wikipedia as a source of information and as a proxy for measuring the global attention to real-world events has been studied [24, 38, 55, 56]. The results of our study add further evidence of the value of Wikipedia data in the field of digital epidemiology, especially for capturing information seeking behavior, and attention patterns during disease outbreaks [57].

We showed Wikipedia data can capture collective attention during outbreaks, however, we did not link such signal with a measure of individual behavioral response or the adoption of health protecting behaviors in the population. Detecting health-related behavioral changes from Web sources remains a challenging task. Previous studies have used TV viewing data to infer the behavioral response during the 2009 A/H1N1 pandemic in Mexico [58]. More recently, Poletto el al. [35] showed that an increased collective attention was correlated to

changes in the hospital management of MERS-Cov patients, reducing the time from admission to isolation. Further research is needed to infer causal patterns between collective attention and behavioral responses, and to identify the most suitable approach to integrate them into disease-behavior models.

## Materials and methods

### Data sources

**Wikipedia pageview counts.** We collected hourly pageview data of the English Wikipedia pages "Zika virus" (https://en.wikipedia.org/wiki/Zika_virus) and "Zika fever" (https://en.wikipedia.org/wiki/Zika_fever) and their counterparts in 96 different Wikipedia projects. The complete list of the 128 monitored Wikipedia pages is provided in S1 Table of the Supporting Information.

The "Zika virus" and "Zika fever" pages are the only two pages in the English Wikipedia that provide information on the disease (note that "Zika" redirects to "Zika fever") and on the pathogen causing the disease ("Zika virus"). They were, by far, the most accessed pages among all Zika related articles in the English Wikipedia, with the "Zika virus" page totalling almost 8 million worldwide views in 2016, and about 800,000 worldwide pageviews for "Zika fever".

While aggregate hourly and daily pageview data for Wikipedia articles by language is released by the Wikimedia Foundation in the form of data dumps (https://dumps.wikimedia.org/other/pageviews/readme.html) and APIs (https://wikitech.wikimedia.org/wiki/Analytics/AQS/Pageviews), the geographic breakdown of this data is not made publicly available due to privacy reasons. The Wikimedia Foundation discards raw traffic data after a short retention window, but it collects and retains aggregate historical pageview counts with a geographic breakdown, dating back to 2015 (https://wikitech.wikimedia.org/wiki/Analytics/Data_Lake/Traffic/Pageview_hourly). The pageview data with geographical aggregations used in this study provides the total view counts and the following information for each page: hour, day, year, city, subdivision, country. Geo-location is based standard industry methods which provide a 90% accuracy at state level in the US and a 86% accuracy for cities in the US within a 50 km radius (https://wikitech.wikimedia.org/wiki/Analytics/Systems/Cluster/Geolocation).

Access to this nonpublic pageview data was granted from the Wikimedia Foundation under a non-disclosure agreement as part of its formal collaboration policy. For the analysis conducted in this study, we first selected the pageview counts that were localized in the United States only, from January 1, 2016 until December 31, 2016. We then aggregated all pageview counts for the 128 monitored Wikipedia pages at daily and weekly timescale.

**Web news.** Data were downloaded from the Global Database of Events, Language and Tone (GDELT—http://www.gdeltproject.org), available on the Google Cloud Platform. The GDELT is created from real-time translation of worldwide news into 65 languages and updated every 15 minutes. Whenever GDELT detects a news report breaking anywhere the world, the report is then translated, processed to identify all events, counts, quotes, people, organisations, locations, themes, emotions, relevant imagery, video, and embedded social media posts. All the information is made available through an API.

In our study, we collected all news items published online in 2016, which mentioned the words "Zika" and "United States", through the Google Cloud Platform. More specifically we selected those items matching *TAX_DISEASE_ZIKA* as a Theme and *United_States* as a Location. Themes (*V2Themes*) and Locations (*V2Locations*) are automatically identified by the platform based on the textual content of the items and each item can be assigned multiple themes or locations. The complete query is provided in the S1 Appendix of the Supporting Information. The dataset contains a total of 112,706 news items from 7,737 different Web

news outlets, in any of the 65 languages covered by the GDELT platform. News outlets are not necessarily based in the United States, the only constraint is that each news item mentions the United States. Time series analysed in our study report the number of news items citing Zika that appeared each day in 2016. Multiple mentions of the word "Zika" in the same article are not counted, therefore we consider the volume of individual articles mentioning Zika. Meta-data associated to each news item allow to select only news mentioning a specific geographic entity beyond the United States, such as States or counties. Each item can be associated to multiple States, therefore in the State level analysis the same item can be counted in the time series of different States at the same time.

**TV captions.**   Data were downloaded from the TV News Archive (https://archive.org/details/tv) which is a research library service launched in September 2012. The service is provided by the Internet Archive which, among other sources, collects and preserves television news. The TV News Archive repurposes closed captioning to enable users to search, quote and borrow U.S. TV news programs. For this study, we collected TV news items by searching all mentions of the word "Zika" in the closed captions of any TV News show aired in the United States in 2016, available from the Archive. For each item, the following information is provided: time, TV station, TV program, text snippet of the caption. In total, the dataset comprises 23,855 timestamped mentions of the word "Zika" from 1,410 different TV programs, both in English and Spanish, aired by 64 U.S. TV stations. We did not limit our query to those languages, since the TV News Archive includes broadcasting networks in other languages too. However, English and Spanish were the only languages resulting from our search. Time series analyzed in our study report the number of mentions of "Zika" that appeared each day in 2016, thus including multiple mentions of the word in the same program.

**Zika case notification data.**   Incidence data of the Zika virus in the United States was collected from the weekly reports published by the CDC. The reports and the associated data were made publicly available by the CDC on GitHub (https://github.com/cdcepi/zika). The CDC epidemiological reports provide the cumulative number of Zika cases by State, starting from February 24, 2016. Additional case counts of January-February 2016 were extracted from CDC official media releases and included in the dataset.

## Vector Autoregression models and Granger causality test

We build the Vector Autoregression models using the R package VARS [29]. The VAR model is of the following form:

$$\mathbf{y}_t = A_1 \mathbf{y}_{t-1} + \cdots + A_L \mathbf{y}_{t-L} + CD_t + TD_t + \mathbf{u}_t \tag{1}$$

where $\mathbf{y}$ is the vector of endogenous variables, which has dimension $3 \times 1$ for daily time series (including Wikipedia pageviews, Web news, TV captions) and $4 \times 1$ for weekly time series (with the addition of ZIKV incidence). $L$ is the lag order and $\mathbf{u}$ assigns a spherical disturbance term of the same dimension of $\mathbf{y}$. The model also includes both a constant and a trend regressors represented by $CD_t$ and $TD_t$. The model is fit by OLS per equation. We determine the optimal lag length $L$ by comparing the Schwartz-Bayes information criterion (BIC) and the Hannan-Quinn criterion for lags up to 40 days for daily time series and up to 8 weeks for weekly time series.

Using the R package VARS, we perform two causality tests for both models, at daily and weekly scale, testing the causal hypothesis for each variable in the model. The first test is a F-type Granger-causality test. The second is a Wald-type test that is characterized by testing for nonzero correlation between the error processes of the cause and effect variables.

## Equal-time regression model

We model the weekly number of pageview counts to Zika-related Wikipedia pages in each state with a linear regression of the form

$$\hat{PV}_s(w) = \sum_i^n b_i X_i \tag{2}$$

where $\hat{PV}_s(w)$ is the Wikipedia pageview count in state $s$ on week $w$, rescaled by the state population. The rescaling of pageview data takes the form:

$$\hat{PV}_s(w) \propto N_s^\beta \tag{3}$$

where $N_s$ is the state population and $\beta = 1.1397$ is a scaling exponent independently estimated on the total volume of pageviews in each state by adopting the probabilistic framework of Leitão et al.[59] (details are reported in the S2 Appendix of the Supporting Information). By K-fold ($k = 10$) and leave-one-out cross validation, we test the performance of the model considering different linear combinations of features $X_i$. Specifically, we considered as model features the weekly media timelines $Y(w)$, where $Y = $ TV, Web or Web$_{state}$, and Web$_{state}$, and Web$_{state}$ represents the selection of Web news mentioning only a specific state name together with the word "Zika". We also consider as a reference the case $Y(w) = $ ZIKV$_s(w)$, where ZIKV$_s(w)$ is the weekly number of reported ZIKV cases in state $s$. To take into account the saturation effect due to media exposure, we also considered an exponentially decaying function of the media timelines $Y$ ($Y = $ TV, Web):

$$m(Y) = \sum_{\Delta t=1}^{\Delta t_{max}} e^{-\frac{\Delta t}{\tau}} Y(w - \Delta t) \tag{4}$$

where $\tau$ is a free parameter, setting the memory time scale, and $\Delta t_{max}$ is defined by the total length of the time series up to week $w$ ($\Delta t_{max} = w$). Thus, the full model with all the 5 media features under consideration takes the following form:

$$\hat{PV}_s(w) = a \cdot \text{TV}(w) + b \cdot \text{Web}(w) + c \cdot m(\text{Web}) + d \cdot m(\text{TV}) + e \cdot \text{Web}_{state}(w). \tag{5}$$

A list of the best estimates for the model's coefficients and the 10 feature combinations of Table 1 is reported in the S6 Table.

## Supporting information

**S1 Appendix. GDELT query.** SQL code has been used to query the GDELT platform through the Google BigQuery API.
(PDF)

**S2 Appendix. Wikipedia pageview scaling.**
(PDF)

**S1 Table. Wikipedia pages under study.** Full list of the 128 Wikipedia pages whose page view counts were monitored in the study. The field language refers to the language codes defined by ISO 639-1 and ISO 639-3.
(PDF)

**S2 Table. Correlations between Wikipedia pageviews, the Web news mentioning Zika and TV close captions in 2016.** The table reports the Pearson's correlation coefficient $r$ for the Wikipedia page view counts, the Web news mentioning Zika and the TV close captions at

national level. All values of $r$ are statistically significant at $p < 10^{-4}$.
(PDF)

**S3 Table. Correlations between Wikipedia pageviews and news mentioning Zika by state.**
All states are ranked by Pearson's $r$ values, in descending order.
(PDF)

**S4 Table. Correlations between Wikipedia pageviews and ZIKV incidence by state.** All
states are ranked by Pearson's $r$ values, in descending order.
(PDF)

**S5 Table. Correlations between news mentioning Zika and ZIKV incidence by state.** All
states are ranked by Pearson's $r$ values, in descending order.
(PDF)

**S6 Table. Parameters of the best fitted models for all selected features.**
(PDF)

**S7 Table. Comparison of model performance for 49 states and D.C.**
(PDF)

## Acknowledgments

We gratefully acknowledge the Wikimedia Foundation for supporting this work through their
formal collaboration and open access policies. We thank Dario Taraborelli for his early sup-
port of this study.

## Author Contributions

**Conceptualization:** Michele Tizzoni, André Panisson, Daniela Paolotti, Ciro Cattuto.

**Data curation:** Michele Tizzoni, André Panisson, Ciro Cattuto.

**Formal analysis:** Michele Tizzoni, André Panisson, Ciro Cattuto.

**Investigation:** Michele Tizzoni, André Panisson, Daniela Paolotti, Ciro Cattuto.

**Methodology:** Michele Tizzoni, André Panisson, Daniela Paolotti, Ciro Cattuto.

**Software:** Michele Tizzoni, André Panisson.

**Supervision:** Ciro Cattuto.

**Visualization:** Michele Tizzoni.

**Writing – original draft:** Michele Tizzoni.

**Writing – review & editing:** Michele Tizzoni, Daniela Paolotti, Ciro Cattuto.

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
