## [Decision Letter · Decision Letter 0]

12 Aug 2019

Dear Dr Tizzoni,

Thank you very much for submitting your manuscript 'The impact of news exposure on collective attention during epidemics: case study of the 2016 U.S. Zika outbreak' for review by PLOS Computational Biology. Your manuscript has been fully evaluated by the PLOS Computational Biology editorial team and in this case also by independent peer reviewers. The reviewers appreciated the attention to an important problem, but raised some substantial concerns about the manuscript as it currently stands. While your manuscript cannot be accepted in its present form, we are willing to consider a revised version in which the issues raised by the reviewers have been adequately addressed. We cannot, of course, promise publication at that time.

Sincerely,

Matthew (Matt) Ferrari

Associate Editor

PLOS Computational Biology

Rob De Boer

Deputy Editor

PLOS Computational Biology

[LINK]

The reviewers were positive in their reviews, but raised a number of issues that should be addressed in revision. All 3 reviewers, in some form, raised the issue of interpreting correlations as causative and the direction of that causation. R3, in particular, raises several issues about whether or not information seeking or simply increased attention, and specifically increased media attention. This should be discussed further in a revision.

R1 and R3 raise several questions about the inclusion of articles in the study (and the possibility of duplicates). If it is possible to address these by looking at the sensitivity of the results to alternate inclusion criteria, this should be done. If this is not possible, then the authors should at least provide text that would answer to the reviewers' questions in the main text.

The reviewers also raise a number of small technical questions that should be clarified.

Reviewer's Responses to Questions

**Comments to the Authors:**

Reviewer #1: The authors have submitted an enjoyable, thoughtful, and timely paper. I enjoyed assessing it. The manuscript and the study harbor some important weaknesses, however, that should be addressed before moving forward.

Although the paper is generally well written, the piece would benefit from some additional copyediting. For example, there appears to be an errant question mark in brackets on page 11 of the PDF. The title also may be somewhat misleading, as I believe this is a study of behavior in the US relative to the Zika outbreak but the 2016 outbreak wasn't limited to the US nor was it really centered in the US, per se. The transmission that actually occurred in the US was quite limited. The authors might just want to revisit how exactly they are framing the phenomenon being studied. There was US transmission but much of the news coverage driving the search behavior reflected international events, unless I am mistaken.

The authors also miss an opportunity to connect this work to other highly relevant results that have appeared in other journals. Consider, for example, Southwell et al. in Emerging Infectious Diseases, a piece that connects news coverage, search data, and social media data regarding the Zika outbreak discussed here. The story is remarkably similar to what we found and so that would be an important foundational citation for the discussion here. Of course, the outcome measures of information seeking are somewhat different but nonetheless the story of ephemeral effect driven by news coverage that was not completely tied to actual epidemiological patterns is consistent. Beyond that, the authors also have an opportunity to connect this piece with a limited but nonetheless important subset of the communication research literature which has looked at communication as a process unfolding over time and which as a result has included time-based analysis of news effects on behavior.

On a related note, the other major limitation of the paper is the mismatch between theoretically longitudinal concerns with correlational data. Chances are good that time series analysis will tell us a similar story as the correlational analyses here but nonetheless because cases are ordered in time there is a risk that the analysis presented unfairly capitalizes on lurking autocorrelation in the data. The authors should comment on that and either present their findings using time series analysis or at least assure readers that such analysis provides a similar story in a detailed note.

If the authors can address these concerns, they will very likely improve the contribution of the piece.

Reference

Southwell, B. G., Dolina, S., Jimenez-Magdaleno, K., Squiers, L. B., & Kelly, B. J. (2016). Zika virus–related news coverage and online behavior, United States, Guatemala, and Brazil. Emerging Infectious Diseases, 22(7), 1320-1321.

Reviewer #2: This paper attempts to measure the drivers leading to information seeking in Wikipedia during the 2016 U.S. Zika outbreak. Specifically, the authors applied various statistical approaches to compare Wikipedia page view data against U.S. news outlets and TV shows at various spatial resolutions (i.e., national, state, and city level). The authors found that Wikipedia searches were driven by national media coverage and not by the magnitude of the outbreak. Although the methodology used is simple, the results are important for understanding the impact of media exposure on collective population behavior.

Major Comments

• Overall, very limited results are presented for the state and city-level analyses. Given the geo-localized Wikipedia pageview available to the authors, the paper could be strengthened by providing timeseries at the state and city level as well as more detailed analyses that showcase the various behavioral responses by geography and language.

• The authors considered daily pageviews on 128 different Zika-related Wikipedia articles in 96 languages. However, it is not clear if the Web news captured all 96 languages. Similarly, the TV captions appear to only include English and Spanish. Therefore, it is not clear if the analyses performed are correct given that the different data streams analyzed may be measuring different populations. In short, the same languages should be captured in all the data streams.

• A discussion on why the authors decided to only use Zika and Zika virus for most of the articles analyzed is warranted. Similarly, a justification on why they decided to focus on 43 states with population in excess of 1 million is needed.

• Are duplicate news articles included or removed from the analyses (GDELT)?

• What is the scaling exponent for city-level analyses? Does the scaling exponent (beta) change for each state or it remains constant across all the states?

• The authors should consider including tables for the correlations between Wikipedia and TV shows in the supplementary material.

• It’s hard to read the states in Figure 3, consider increasing the font.

Minor Comments:

• A reference is missing in page 11.

Reviewer #3: This is a very important, well-articulated, and thorough study, creatively using preexisting novel data to evaluate the relationships between Zika cases, news coverage, and Wikipedia visits. I have several reservations regarding the analysis and would strongly suggest you adding some missing literature (see below), but I believe these may be addressed through a revision. I applaud the authors for the meticulous and creative methodological work. My specific comments are as follow:

1) I suggest you add Sandman's perspective of risk into your literature and into the interpretation of the results. Sandman in his work on risk perception did a pretty good job explaining why experts perceive risk differently from health organizations such as the CDC. He has a list of what he called outrage factors - like novelty (see Sandman, 1987. Risk Communication: Facing Public Outrage), that could really explain why people in your study seem to have lost interest with time. Paul Slovic had some similar work that could also be implemented here - in any case, I think you should add a discussion of risk perception of laypersons and how it is different from that of experts - both Sandman and Slovic could be useful here. Following their perspective - your results are expected and make sense.

2) Information seeking - you seem to shift back and forth between collective attention and information seeking. First, I think some additional literature is needed re health info seeking - Nehama Lewis recently wrote a value on information seeking and scanning (2017) for the International Encyclopedia of Media Effects that could be useful. Second, be consistent with your main DV. Do you measure attention to information seeking?

3) Your analysis will also benefit from discussing a recent study that looked at the relationships between Zika news coverage and public knowledge, familiarity and information sharing in the US - using a large national survey. The paper is: Ophir & Jamieson (2018). The Effects of Zika Virus Risk Coverage on Familiarity, Knowledge and Behavior in the U.S. – A Time Series Analysis Combining Content Analysis and a Nationally Representative Survey. This seems especially relevant to your inquiry.

4) on p3 - you need to elaborate more on why ZIKV is a communication challenge - the explanations you bring are for why it was a public health challenge, not necessarily a communication one. Use risk or crisis communication literature here.

5) p4 - CDC should be Centers for Disease Control and Prevention (add "and Prevention")

6) p5 - I'm very concerned with the use of Pearson correlation for variables that will definitely be influenced by autocorrelation. In such case, where I expect autoregression in both observed and unobserved variables that could be correlated with the dependent, I strongly prefer the use of vector auroregression models (VARS). VARS models could be coupled with Granger causality tests to support a causal direction as well.

7) also on p5 - it has to do with the literature but I really don't think you should have expected a linear relationship between number of new cases and public reaction. First of all, while it's true that new cases were added, other prior cases were solved... and in almost all cases - Zika came and went without leaving any harm. So it completely makes sense to me that people lost interest in Zika after a while (with an exception for the Rio olympics where people probably considered its effects on visitors and athletes-- which was also prevalent in news coverage).

8) Similarly for the media - we have strong reasons to expect the media NOT to follow the number of cases. Again, I think some literature is missing here - something on newsworthiness (Galtung & Rouge, 1965 or something similar). For various reasons, Zika was just not interesting for journalists. It gained some attention when it was new and mostly unknown, but other than its effects on golfers etc. it wasn't a big deal for journalists. You can see more about changes in content in Ophir. (2018). Coverage of epidemics in American newspapers through the lens of the Crisis and Emergency Risk Communication Framework

9) p6 - how reliable do you believe the geographic data is? is it based on ip addresses? these are often inaccurate

10) still on p6 - you measure correlations, but by doing that you assume the public follows the media. While plausible, it could also be that journalists "feel" the public interest and opinion (for example in social media) and are therefore affected by people's attention to the disease. You should be careful about the causal assumption here and explain that it could be the other way around. Again - it would be helpful to use autoregressive models with Granger causality to provide additional support for direction, including the optimal lag.

11) p11 - line 254 - there is a question mark where a citation number should appear - "disease similar to ZIKV [?]"

12) p12 - I think you're too harsh on yourself saying you didn't measure public behavior. Information seeking is a behavior - you could say that you didn't measure behavior on the individual level, or did not measure health-related behavior, but information seeking is definitely a behavior and Wikipedia seems like a reasonable proxy for that.

13) I'm worried about your decision to limit articles to those including both Zika and United States - Do you really think all relevant articles will include the term "United States"? For example,- people interested in the Rio olympics in Brazil will look at articles that do not use the term US. Also - you assume that only articles that explicitly connect the disease to the States will have an impact, which might not be the case. Anyway - I would remove the US condition from your search and look at all US media mentions of Zika. But if you decide to stick with your decision, at least explain why you did so in the discussion section and how it affects your conclusions.

14) notice that figures are in low resolution. Please provide sharper versions in the revision (e.g., hard to read states' names, etc).

**Have all data underlying the figures and results presented in the manuscript been provided?**

Reviewer #1: Yes

Reviewer #2: No: The Wikipedia page view was provided under a non-disclosure agreement.

Reviewer #3: No: The authors stated some data will not become available due to restrictions. They did, however provide the correlation matrix upon which the heatmap was built

PLOS authors have the option to publish the peer review history of their article (what does this mean?). If published, this will include your full peer review and any attached files.

Reviewer #1: Yes: Brian Southwell

Reviewer #2: No

Reviewer #3: No

---

## [Decision Letter · Decision Letter 1]

6 Jan 2020

Dear Dr Tizzoni,

We are pleased to inform you that your manuscript 'The impact of news exposure on collective attention in the United States during the 2016 Zika epidemic' has been provisionally accepted for publication in PLOS Computational Biology.

Both reviewers were appreciative of the efforts made by the authors to address their concerns. Reviewer 3 did make a few small suggestions to language. I would encourage the authors to consider these changes as the suggested language may make arguments more acceptable to readers. These changes can be made at the proofs stage and I see no reason to delay the editorial process any further. 

In the meantime, please log into Editorial Manager at https://www.editorialmanager.com/pcompbiol/, click the "Update My Information" link at the top of the page, and update your user information to ensure an efficient production and billing process.

One of the goals of PLOS is to make science accessible to educators and the public. PLOS staff issue occasional press releases and make early versions of PLOS Computational Biology articles available to science writers and journalists. PLOS staff also collaborate with Communication and Public Information Offices and would be happy to work with the relevant people at your institution or funding agency. If your institution or funding agency is interested in promoting your findings, please ask them to coordinate their releases with PLOS (contact ploscompbiol@plos.org).

Thank you again for supporting Open Access publishing. We look forward to publishing your paper in PLOS Computational Biology.

Sincerely,

Matthew (Matt) Ferrari

Associate Editor

PLOS Computational Biology

Rob De Boer

Deputy Editor

PLOS Computational Biology

Reviewer's Responses to Questions

**Comments to the Authors:**

Reviewer #1: The authors have addressed earlier review comments adequately. Thank you for the effort.

Reviewer #3: The authors took my and other reviewers' comments seriously and improved the paper accordingly. The result is a more careful analysis that takes into consideration the methodological and theoretical points raised by the reviewers, while better connecting to recent findings in other studies. I appreciate the hard work given to this manuscript in its original and revised versions. I believe the revised version is of considerable theoretical and practical importance for the fields of health, risk, and crisis communication.

I have very minor suggestions for the authors, but believe these could be communicated directly with the editor and do not require another round of peer-review:

1) remove the word "always" from line 212 on page 9

2) In line 297 of page 12, change the term "we could prove" to "our data support" (or at least change prove to support)

3) p13 - change "associated to the disease" to "associated with the disease"

4) p13, line 305 - change "may be obvious" to "may be expected"

5) notice Ophir & Jamieson (2018) was published in Volume 35 Issue 1 of Health Communication, pages 35-45 (https://www.tandfonline.com/doi/full/10.1080/10410236.2018.1536958)

6) I would find a more cautious language for "social media provide an amplification channel for traditional media sources"-- research on the topic is not as consistent as this sentence suggest (e.g., Harder et al., 2017. Intermedia agenda setting in the social media age: https://journals.sagepub.com/doi/10.1177/1940161217704969)

Once again, I commend the authors' meticulous and rigor work and look forward to reading the final publication.

**Have all data underlying the figures and results presented in the manuscript been provided?**

Reviewer #1: Yes

Reviewer #3: Yes

PLOS authors have the option to publish the peer review history of their article (what does this mean?). If published, this will include your full peer review and any attached files.

Reviewer #1: Yes: Brian Southwell

Reviewer #3: Yes: Yotam Ophir, Ph.D.

---

## [Editor Report · Acceptance letter]

19 Feb 2020

PCOMPBIOL-D-19-00953R1 

The impact of news exposure on collective attention in the United States during the 2016 Zika epidemic

Dear Dr Tizzoni,

I am pleased to inform you that your manuscript has been formally accepted for publication in PLOS Computational Biology. Your manuscript is now with our production department and you will be notified of the publication date in due course.

With kind regards,

Matt Lyles
